# Cryo-EM structures of Banna virus in multiple states reveal stepwise detachment of viral spikes

Zhiqiang Li[1,2,3], Han Xia [1,3], Guibo Rao[1,3], Yan Fu[1], Tingting Chong[1,2], Kexing Tian[1,2], Zhiming Yuan [1] ✉ & Sheng Cao [1] ✉

Banna virus (BAV) is the prototype *Seadornavirus*, a class of reoviruses for which there has been little structural study. Here, we report atomic cryo-EM structures of three states of BAV virions—surrounded by 120 spikes (full virions), 60 spikes (partial virions), or no spikes (cores). BAV cores are double-layered particles similar to the cores of other non-turreted reoviruses, except for an additional protein component in the outer capsid shell, VP10. VP10 was identified to be a cementing protein that plays a pivotal role in the assembly of BAV virions by directly interacting with VP2 (inner capsid), VP8 (outer capsid), and VP4 (spike). Viral spikes (VP4/VP9 heterohexamers) are situated on top of VP10 molecules in full or partial virions. Asymmetrical electrostatic interactions between VP10 monomers and VP4 trimers are disrupted by high pH treatment, which is thus a simple way to produce BAV cores. Low pH treatment of BAV virions removes only the flexible receptor binding protein VP9 and triggers significant conformational changes in the membrane penetration protein VP4. BAV virions adopt distinct spatial organization of their surface proteins compared with other well-studied reoviruses, suggesting that BAV may have a unique mechanism of penetration of cellular endomembranes.

Members of the order *Reovirales* (formerly the family *Reoviridae*) are non-enveloped icosahedral viruses with a genome consisting of 9–12 segments of double-stranded RNA (dsRNA)[1]. The structures of reoviruses, including prototype species of the genera *Aquareovirus*, *Cypovirus*, *Orbivirus*, *Orthoreovirus*, *Phytoreovirus* and *Rotavirus*, have been studied by X-ray crystallography or by cryo-electron microscopy (cryo-EM)[2–7]. The intact reovirus particles are composed of one, two or three concentric capsid shells. Reoviruses have a common process of uncoating their surface proteins during the infection of cells, after which the remaining part of the viral particle (termed the core) is released into the cytoplasm of the host cell. The reoviral cores are relatively stable nanostructures for endogenous transcription of viral mRNA, protecting the viral genomic dsRNA from antiviral strategies of the host cell. The core of cytoplasmic polyhedrosis virus (genus

*Cypovirus*) contains only one icosahedral capsid shell[6], but most studied reoviral cores are double-layered particles (DLPs). Based on the presence or absence of turret structures on the core surface at the five-fold vertices, reoviruses are divided into two families, turreted *Spinareoviridae* (e.g., orthoreoviruses, cypoviruses, aquareoviruses), and relatively smooth nonturreted *Sedoreoviridae* (e.g., rotaviruses, orbiviruses, phytoreoviruses)[8,9].

The surface proteins of reoviruses can be functionally classified into two groups, one for receptor binding and one for membrane penetration. While the core structures are relatively conserved among sedoreoviruses, the arrangement of surface proteins shows large variation from virus to virus. Recently, rhesus rotavirus (RRV)[10] and bluetongue virus (BTV, *Orbivirus*)[11] were studied structurally, which shed some light on the molecular mechanisms underlying membrane

[1] CAS Key Laboratory of Special Pathogens, Wuhan Institute of Virology, Center for Biosafety Mega-Science, Chinese Academy of Sciences, Wuhan 430071, PR China. [2] University of Chinese Academy of Sciences, Beijing 100049, PR China. [3] These authors contributed equally: Zhiqiang Li, Han Xia, Guibo Rao. ✉e-mail: yzm@wh.iov.cn; caosheng@wh.iov.cn

penetration by non-enveloped viruses[12,13]; the membrane fusion mechanisms have been studied extensively for many enveloped viruses[14,15]. On rotaviruses, during membrane perforation, 60 spikes (VP4) are proteolytically cleaved into VP8* (receptor binding) and VP5* (membrane penetration), which triggers structural rearrangement in VP5* from an 'upright' to a 'reversed' conformation[10]. On the outer layer of BTV virions in contrast, there are 120 VP5 trimers for membrane penetration[16]. Upon exposure to low pH, the unfurling domains of BTV VP5 refold into a six-helix stalk, which is then inserted into the target membrane to mediate membrane penetration, and a surface loop in the anchoring domain converts into a β-hairpin to anchor VP5 on the virus core[11].

Banna virus (BAV), containing a genome of 12 segments of dsRNA, is the prototype species of the genus *Seadornavirus* (derived from *South-east Asian dodeca RNA virus*) in the family *Sedoreoviridae*[17]. BAV was first isolated in 1987 from the cerebrospinal fluid of a patient suffering from encephalitis in southern China[18]. Subsequently, BAV has been isolated from various animals, such as mosquitoes, midges, and livestock in China, Vietnam, and Indonesia; it is considered to pose a possible threat to animals and humans[19]. BAVs are thought to be transmitted by mosquitoes[20], from which some dsRNA viruses have been isolated with antigenic properties close to those of the original BAV[21]. Intact BAV has seven structural proteins (VP1, VP2, VP3, VP4, VP8, VP9, and VP10), while its core contains five proteins (VP1, VP2, VP3, VP8, and VP10)[22]. Among these structural proteins, only one crystal structure has been solved so far for receptor binding protein VP9, which has a trimeric conformation in solution and consists of an *N*-terminal stalk region and a *C*-terminal head domain[23]. Because of the lack of high-resolution structures of BAV, it remains elusive how the viral proteins are organized in the virion and how they are involved in cell entry.

In this study, a BAV strain that does not replicate in vertebrate cell lines was used for cryo-EM single particle analysis. Our structural studies reveal that although BAV shares some similarities with other sedoreoviruses, especially BTV, unique composition and conformation of viral spikes require this virus to take different steps to enter the cell.

## Results

### Infectivity of BAV strain YN15-126-01
Currently BAVs can be classified into three groups, A, B, and C, based on phylogenetic analysis of the complete coding sequence (CDS) of segment 12[24]. The CDS of segment 12 of BAV strain YN15-126-01 has high nucleotide identity with other reported group A BAVs [YNV/01-1 (95.82%), SC043 (95.83%), and BAV-Ch (95.82%)] isolated in Yunnan, China; the amino acid identities of YN15-126-01 with YNV/01-1, SC043, and BAV-Ch are 95.65%, 96.14%, and 96.62%, respectively.

To evaluate the infectivity of BAV YN15-126-01 in a variety of host cells, six vertebrate cell lines (LMH, BHK-21, PK15, Vero, A549, Huh-7) and one mosquito cell line (C6/36) were inoculated with virus at ratio of copy number of viral RNA to number of cells (R/C) = 1 or 1000. Viral growth kinetics analysis of the supernatant of the infected C6/36 cells indicated that BAV YN15-126-01 replicated efficiently, and could reach up to $10^7$ copies/ml. No viral growth was detected in the cell lines derived from vertebrates (Supplementary Fig. 1a). In addition, BAV caused a typical cytopathic effect (CPE), characterized by shrinkage and death, in C6/36 cells (Supplementary Fig. 1b), whereas no CPE was observed in the vertebrate cells.

### Structural overview
Visual inspection and 2D class averaging of purified BAV YN15-126-01 virions on cryo-EM micrographs revealed two types of virions: one surrounded by more spikes on the viral surface (hereafter referred to as full particles), and the other with sparser spiky density (partial particles) (Fig. 1a, c and Supplementary Fig. 2a). The ratio of full to partial particles was approximately 3:1. Icosahedral reconstructions of full and partial BAV particles (at ~5.7 Å resolution) revealed that they share overall structural features except for the different number of spikes on the virus surface (Fig. 1b, d). Local reconstructions focused on the fivefold vertices (Supplementary Fig. 3) produced a clear map at 2.6 Å resolution for tracing most residues of VP2, VP4, VP8, and VP10 in the icosahedral asymmetrical unit (ASU) (Fig. 1e–h). Density for VP9 was not well-defined except for the *N*-terminal fragment containing one α-helix. Local reconstruction focused only on VP1 or VP9 produced density maps with sufficient resolution for main chain modeling (Supplementary Fig. 3, 4 and Supplementary Table 1).

In the full and partial BAV virions, genomic dsRNA segments and a number of RNA-dependent RNA polymerases (RdRps, VP1) are packaged in a double-layered icosahedral capsid consisting of three proteins (VP2, VP8, and VP10) (Supplementary Fig. 2c). Immediately inside the inner capsid, RdRp is located near the fivefold vertices. The thinner inner capsid shell is formed by 120 copies of VP2 in two slightly different conformations: VP2A (clustered around the fivefold vertex) and VP2B (around the threefold axis), arranged in a "pseudo T = 2" lattice (Fig. 1e), like in other members of the *Reovirales*. The thicker outer capsid shell is composed of 780 copies of VP8 and 120 copies of VP10, with each VP10 molecule being surrounded by six VP8 trimers (Fig. 1f). Two VP10 molecules are identified in each ASU, VP10A (closer to the icosahedral fivefold axis) and VP10B (closer to the threefold axis). The spikes protruding from the outer capsid shell are heterohexamers composed of a VP4 trimer capped by a VP9 trimer (Fig. 1g). In the full BAV virions, there are 60 spikes sitting on VP10A and VP10B, respectively, while in the partial virions, there are only 60 spikes on VP10A.

### Inner capsid shell
VP2A and VP2B, as chemically identical subunits, share similar overall structures, comprised primarily of three domains: an apical domain located close to the fivefold vertices, a carapace domain, and a dimerization domain, following the nomenclature established for the inner capsid protein of reoviruses (Fig. 1h and Supplementary Fig. 5a, b). The map resolution was sufficient to trace VP2B *N*-terminal tails into nearby VP2 molecules, which strengthens the interactions between adjacent VP2 subunits (Supplementary Fig. 5c, d and Supplementary Table 2). The resolved *N*-terminal sequence (residues 20–174) consists of four loops (L1–L4) and three helices (α1–α3), and L2 (residues 40–116) accounts for ~50% of the *N*-terminal tail (Fig. 2a, b). Five *N*-terminal tails around the fivefold axis form a continuous interaction network (Fig. 2a), likely promoting ordered assembly of icosahedral pentons (pentamers of VP2 dimers) (Fig. 2c, d). In addition, *N*-terminal tails traverse multiple subunits to interact with VP2A and VP2B from neighboring pentons (Supplementary Fig. 5c, d), and thus also contribute to the stability of the whole icosahedral assembly.

### RdRp associated with the inner shell
Asymmetrical reconstruction focused on icosahedral vertices revealed clear density for a non-transcribing RdRp (VP1) molecule near the fivefold pore within the inner capsid shell (Fig. 2e and Supplementary Fig. 6a), and nearby dsRNA is also visible at low contour levels (Supplementary Fig. 6b). Compared with adjacent VP2 subunits in the inner capsid, VP1 shows increased *B*-factors (Supplementary Fig. 6c). Like other reported reovirus RdRps, the *N*-terminal domain and *C*-terminal "bracelet" domain cover the BAV RdRp "fingers–palm–thumb core", forming a cage-like structure with four channels connected to the reaction center (Supplementary Fig. 6d). A structural similarity search using the online DALI server[25] revealed that BAV RdRp is highly similar to other sedoreovirus RdRps despite some local structural variations (Supplementary Fig. 6f). It has been reported that the *N*-termini of multiple inner capsid subunits homologous to VP2A at the icosahedral vertices adopt distinct conformations to hold sedoreovirus RdRps in place[26–28]. Only one *N*-terminal tail of a VP2A subunit interacts with the *C*-terminal bracelet domain of RdRp, near the C5 exit channel (Fig. 2f),

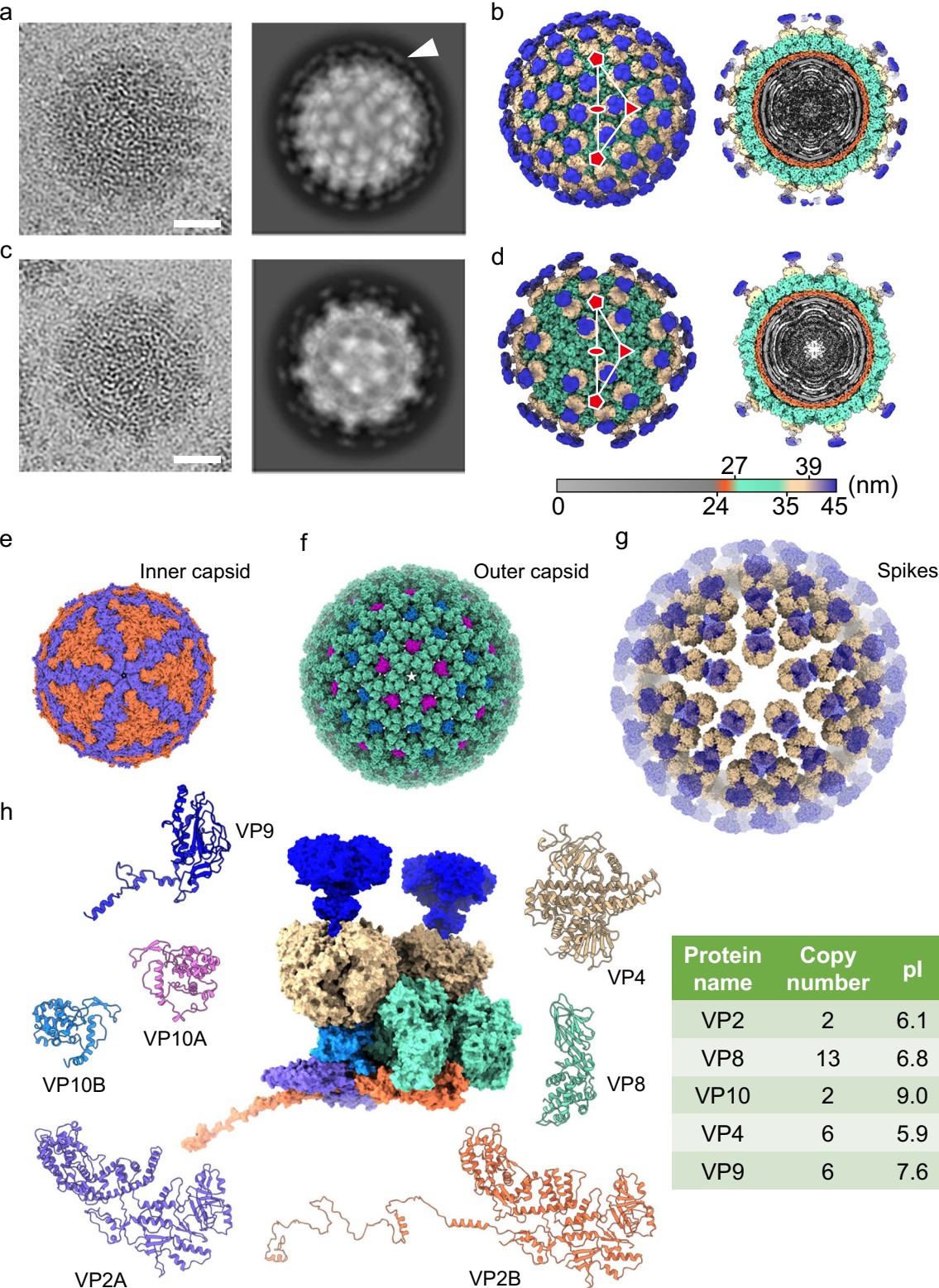

| Protein name | Copy number | pI |
|---|---|---|
| VP2 | 2 | 6.1 |
| VP8 | 13 | 6.8 |
| VP10 | 2 | 9.0 |
| VP4 | 6 | 5.9 |
| VP9 | 6 | 7.6 |

**Fig. 1 | Structural overview of full and partial Banna virions (BAVs).**
**a** Representative 2D image (left) and class averaging (right) of BAV full particles. The spiky density on the viral surface forms a broken ring indicated by an arrow. Scale bar: 20 nm. **b** Surface representation of the full BAV particle viewed along a twofold axis (left) and a cutaway view showing the internal density of the virus (right). The virus is radially colored, showing the viral genome and RdRp (gray), inner capsid shell (orange), outer capsid shell (emerald), and spikes (beige and blue). An asymmetric unit (ASU) is marked by a white triangle. One twofold, one threefold, and two fivefold axes are indicated by an ellipse, a triangle, and two pentagons, respectively. **c** 2D image (left) and class averaging (right) of BAV partial particles. **d** Surface representation (left) and cutaway view (right) of BAV partial particle. **e**–**g** Surface representation of inner capsid shell (**e**), outer capsid shell (**f**), and spikes (**g**) of full BAV particles viewed along a fivefold axis. Maps are presented as solvent-excluded surfaces with a probe radius of 1.4 Å. Proteins are colored as in (**h**). Density for VP9 is shown as a transparent blue surface. **h** Protein composition of an ASU (center, presented as a solvent-excluded surface) of full BAV particles and ribbon diagrams of individual proteins. Copy number and pI values are listed in the table (right).

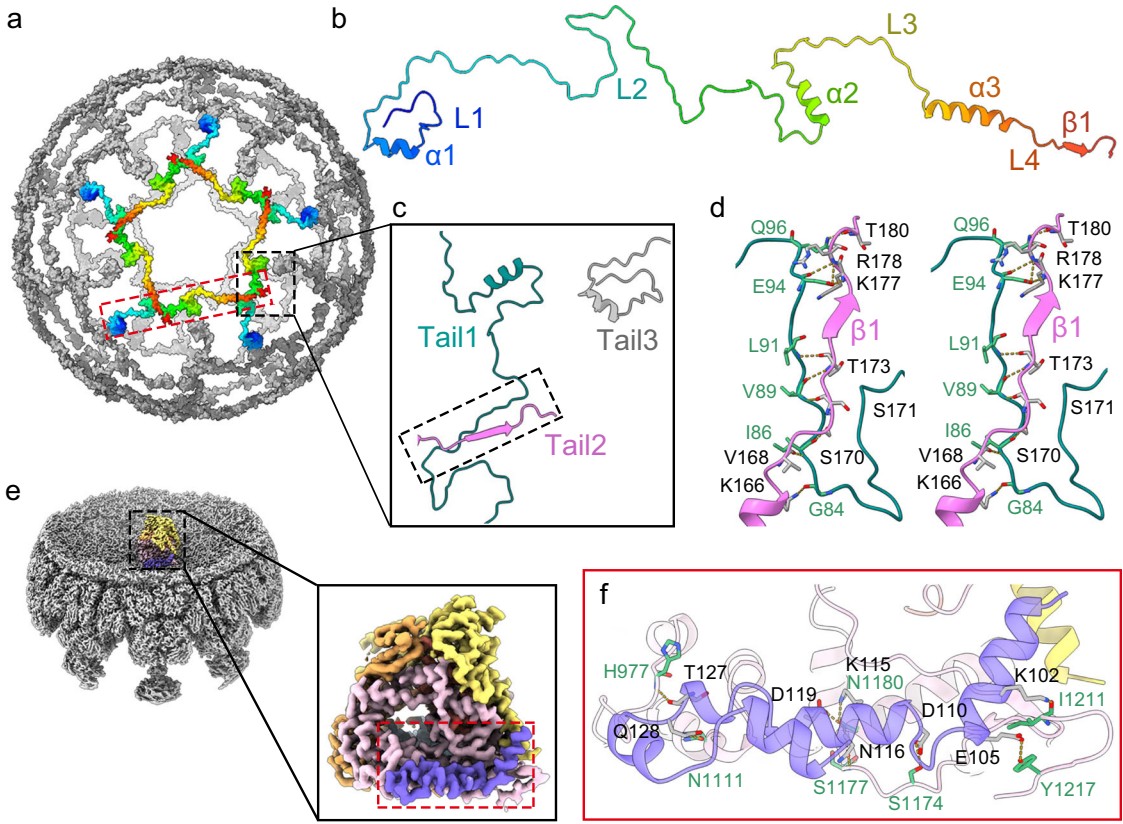

**Fig. 2 | VP2 and VP1 (RdRp). a** Icosahedral network formed by *N*-terminal tails (residues 20–174) of VP2B. The fragments around a fivefold axis are rainbow-colored by residues from the *N*-terminus (blue) to the *C*-terminus (red). To give a full picture of interactions between the tails, the downstream β1 (residues 175–181) are also shown. **b** Zoomed-in view of the red box from **a**, showing an *N*-terminal tail of VP2B, with secondary structural elements labeled. **c** Zoomed-in view of the black box from **a**, showing interactions between two *N*-terminal tails (Tail 1 and Tail 2). Note that although Tail 3 does not directly contact Tail 1, Tail 1 and Tail 3 interact with the same VP2A and VP2B molecules (Supplementary Fig. 5c). **d** Stereo view of the box from **c**, showing hydrogen bonds between Tail 1 and Tail 2. **e** Local reconstruction of RdRp at fivefold vertices. RdRp is color-coded by domain: *N*-terminal domain, yellow; the fingers–palm–thumb core, orange; and the C-terminal "bracelet" domain, pink. The *N*-terminal extension of VP2A interacting with RdRp is colored purple. **f** Zoomed-in view of the red box from **e**, showing the interface between RdRp and the *N*-terminal extension of VP2A. Residues involved in intermolecular hydrogen bonding are labeled and colored green (RdRp) or black (VP2A), respectively.

suggesting that BAV RdRp may be less stable within the inner shell compared with other sedoreovirus RdRps.

## Outer capsid shell

VP8 (780 copies) forms 260 trimers in the outer capsid shell with a thickness of ~80 Å (Fig. 3a). VP8 is arranged in T = 13 icosahedral symmetry, and like its counterparts in other sedoreoviruses[2], each VP8 protomer can be divided into two structural domains: a smaller 'upper' domain (residues 1–7 and 119–227), and a larger 'lower' domain (residues 8–118 and 228–299) (Fig. 3b). The lower domain mainly comprises α-helices and sits on the inner capsid shell, while the upper domain mainly comprises a β-sandwich and is partially exposed on the full or partial virion surface (Fig. 1b, d). Within the trimeric structure, each VP8 molecule wraps with the adjacent subunit in a right-handed manner to bury a surface area of ~1400 Å² on one VP8 molecule. The lattice of VP8 trimers produces 132 channels surrounded by five or six VP8 trimers, which can be classified into three types: I, II and III (Fig. 3a). Type II and III channels are blocked by VP10A and VP10B, respectively (Fig. 3c). VP10A and VP10B interact with all of the six surrounding VP8 trimers (buried surface > 0, Fig. 3c). The type II and III channels are well aligned, except for one VP8 trimer shifting ~10 Å (Supplementary Fig. 7c).

VP10 is the smallest protein (MW: 28.6 kDa) among all the resolved BAV structural proteins, and it also has the highest isoelectric point (pI = 9.0) (Fig. 1h). As revealed by DALI search, no reported structure is similar to that of VP10. The overall shape of VP10 is tetrahedral

(Fig. 3d). VP10 mainly comprises twelve α-helices and four short β-strands (Supplementary Fig. 7a). Superimposition of VP10A onto VP10B revealed that the two conformers are almost identical [root-mean-square deviation (RMSD) of 0.438 Å for 203 Cα atoms]. The maximum deviation is in the *C*-terminal tails located at the tetrahedral vertex near the inner capsid (Fig. 3d). These tails are important for the interactions between VP10 and VP2, resulting in differentially orientated VP10A and VP10B within the outer capsid (Fig. 3e, Supplementary Fig. 7b and Table 3). Within VP10A, a β-hairpin (β1 and β2) and a loop (connecting α11 to α12) on one edge of the tetrahedron insert into the interface between two VP8 trimers (Q and R-trimers) (Fig. 3e). The nearby VP10B molecule inserts into the other side of the interface in a similar way. Hydrogen bonding and salt bridges stabilize these insertions (Supplementary Table 4). Because VP10 interacts with VP2 (inner capsid), VP8 (outer capsid), and VP4 (spikes, see below), VP10 functions as a cementing protein to maintain the integrity of BAV virions.

## VP4: a penetration protein

VP4 trimers are situated in the cavity formed by six VP8 trimers and directly interact with VP10A or VP10B (Fig. 4a). The C3 symmetry is broken for the three *N*-terminal tails (various numbers of residues were traced) within one VP4 trimer, which are involved in forming the asymmetric interface between the VP4 trimer and VP10 monomer (Supplementary Fig. 8a, b). Despite the symmetry mismatch between the VP4 trimer and VP10 monomer, the interface is similar for VP10A and VP10B (Supplementary Table 5). The VP4 trimer has an overall

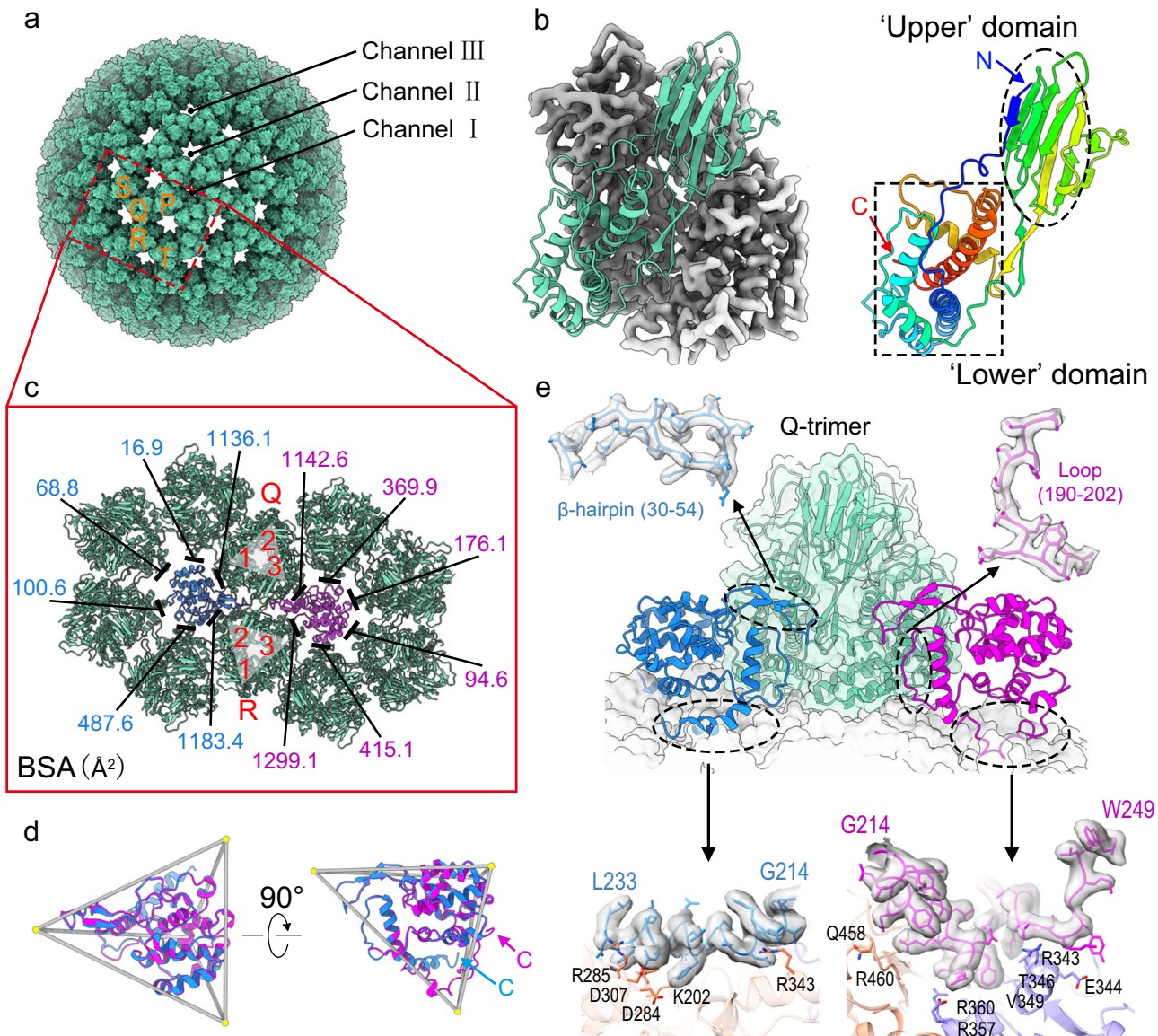

**Fig. 3 | VP8 and VP10. a** Based on the position of VP8 trimers in the ASU, there are five types of VP8 trimers, defined as P, Q, R, S, and T. VP8 arrangement on the outer capsid shell yields three types of channels. The type I channel is located at the icosahedral fivefold vertex. There are 60 type II channels, at the quasi-sixfold axes close to type I channels, and 60 type III channels are situated at the quasi-sixfold position close to icosahedral threefold axes. **b** A VP8 trimer (left) and one VP8 protomer (right). The VP8 protomer is rainbow-colored by residues from the *N*-terminus (blue) to the *C*-terminus (red). **c** Measurement of buried surface area on each VP8 trimer around VP10A (magenta) and VP10B (blue). Part of the VP10 conformer inserts into two neighboring VP8 trimers (labeled the Q- and R-trimers).

Detailed information on the interactions between each protomer in the Q- or R-trimer and VP10A or VP10B is given in Supplementary Table 4. **d** Top view (from outside of the virus, left) and side view (right) of overlayed VP10A (magenta) and VP10B (blue) in a tetrahedral frame. The *C*-termini of VP10A and VP10B are indicated by arrows. **e** Side view of interactions of VP10 conformers with the 'lower' domains of VP8 trimers. For simplicity, only one VP8 trimer (the Q-trimer in **c**) is shown in ribbon representation, superimposed with its cryo-EM density. Two *C*-terminal tails, a β-hairpin, and a loop involved in intermolecular interactions are shown for VP10 in close-up views (superimposed with density).

negatively charged patch near its C3 axis at the VP4−VP10 interface, while positively charged residues R65 and R121 are present on the VP10 side (Fig. 4b, c). At physiological pH, VP4 (pI = 5.9, negatively charged) would electrostatically interact with the basic VP10 protein (pI = 9.0), suggesting that electrostatic interactions between the VP4 trimer and VP10 may play an important role in association of spikes on the virus surface. Binding of VP4 trimers also results in a total ~1800 Å² of buried surface area on VP8 molecules surrounding VP10A, which is ~10% more than for VP8 surrounding VP10B (~1600 Å²), likely contributing to different affinity of VP4 trimers for the outer capsid shell in the full BAV particles.

As revealed by DALI search, part of VP4 is structurally related to the penetration protein VP5 of BTV (Fig. 4d, e), although their primary

structures share no detectable similarity. More specifically, BAV VP4 has structural counterparts to the unfurling and anchoring domains of BTV VP5. Compared with BTV VP5, BAV VP4 has no *N*-terminal dagger domain, but it has a *C*-terminal pedestal domain primarily consisting of a β-sandwich formed by two antiparallel β sheets (β7−β17−β10−β13 and β16−β11−β12−β18). The presence of the unfurling and anchoring domains in BAV VP4 indicates that VP4 may also function as a penetration protein.

### Core structure after alkaline treatment of BAV
To experimentally evaluate the significance of the electrostatic interactions between the VP4 trimer and VP10, we treated BAV virions with alkaline buffer (pH > 10). Negatively stained images of treated particles

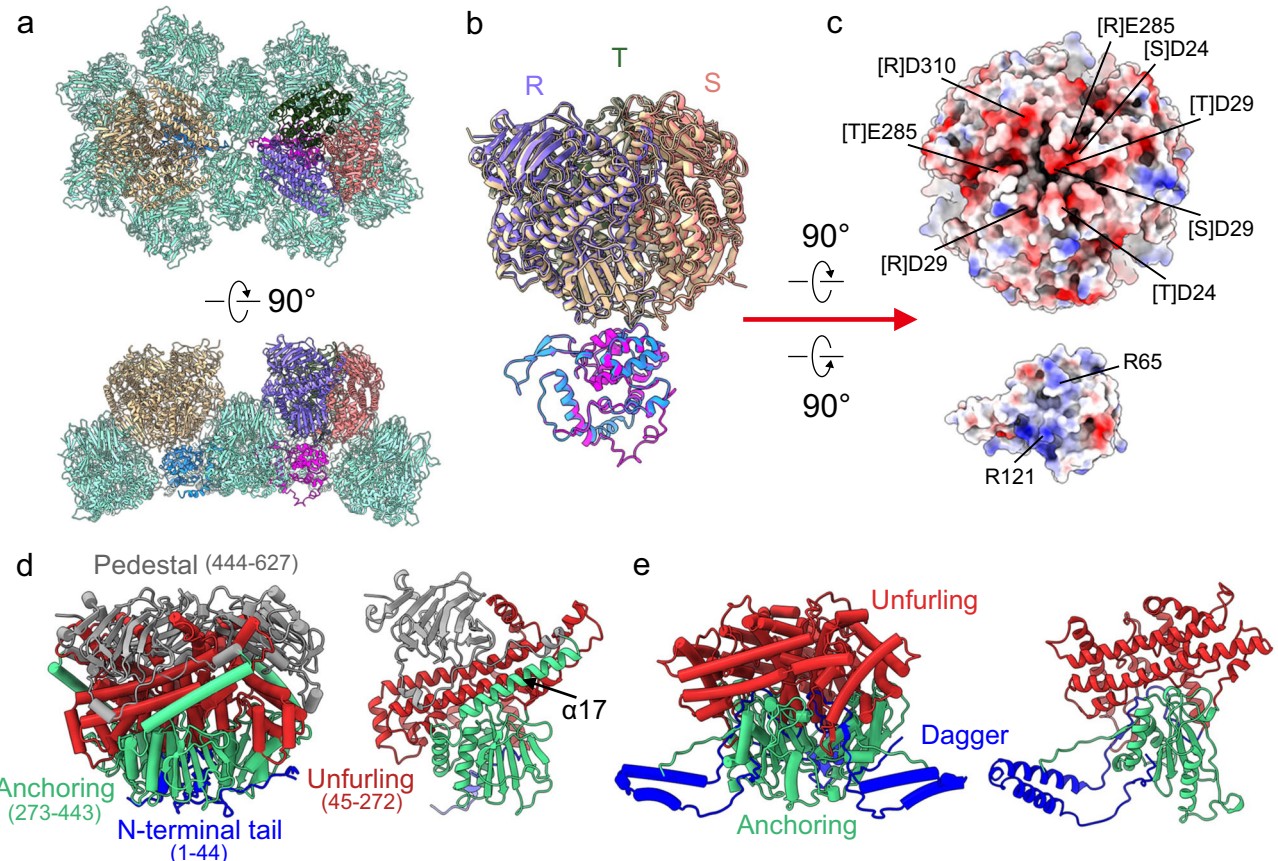

**Fig. 4 | VP4. a** Top and side views of VP4 trimers sitting on VP10A (magenta) and VP10B (blue). The three VP4 protomers on VP10A are colored differently and labeled R, S, and T, respectively. **b** Superimposition of VP4–trimer–VP10A and VP4–trimer–VP10B, revealing that VP4 trimers bind to VP10A and VP10B in a very similar way. **c** The electrostatic potential of the VP4–VP10 interface, ranging from dark blue (most positive) to deep red (most negative). Positively and negatively charged residues are labeled on VP10 and the VP4 trimer, respectively. Note that the interface on the VP4 trimer is symmetry broken (not C3). **d** Side view of the VP4 trimer (left) and one protomer (right). VP4 protomers are colored by *N*-terminal tail (blue), unfurling (red), anchoring (green), and pedestal (gray) domains. Note that α17 of the anchoring domain spans the unfurling domain to connect with the pedestal domain. **e** Side view of the VP5 trimer (based on PDB ID: 3J9E) of BTV (left) and one protomer (right). VP5 protomers are colored as dagger (blue), unfurling (red), and anchoring (green) domains.

showed smooth BAV cores (Fig. 5a right), and compositional analysis by SDS-PAGE revealed that VP4 and VP9 were dissociated from the virions after alkaline treatment (Fig. 5b), suggesting that high pH treatment could indeed disrupt the VP4–VP10 interactions.

Cryo-EM analysis of the BAV core was performed using a similar procedure to that for the full and partial BAV particles. The core particles are DLPs with a diameter of ~70 nm (Fig. 5c, d, Supplementary Fig. 2b, c). The spikes have been completely removed, and the upper domains of VP8 trimers are fully exposed. Accompanying the dissociation of spikes, the fivefold channel for exporting mRNA transcripts has been widened (Supplementary Fig. 9a), and residues near the fivefold axis moved outward (Supplementary Fig. 9a–c). VP10A near the fivefold channel moves more than VP10B (Supplementary Fig. 9d).

To test the transcriptional activity of the core, cell-free transcription reaction was performed in similar conditions to those previously described in ref. 29, and the reaction products were examined using agarose gels on which newly synthesized RNA transcripts were shown to be accumulated in a time-dependent manner (Fig. 5e). Negative-staining TEM showed that there were separately distributed "fibers" surrounding the incubated core particles (Fig. 5f, right panel), also indicating that the alkaline-treated virus particles were transcriptionally active.

### Acidic treatment of BAV particles to remove VP9
The C5 reconstruction focusing on vertices of full particles revealed only low-resolution density for VP9, except the *N*-terminal fragment

containing one α-helix (Fig. 6a), reflecting structural variations in the head domains of the VP9 trimer relative to the *N*-terminal α-helices. The α1-helices from the three protomers form a tripod-like conformation, which is situated in a central cavity surrounded by pedestal domains of the VP4 trimer. Residues from the unfurling domain (D71, S74, and D78) and the pedestal domain (K466, D499, T500, N502, E503, Q525, and K565) of VP4 form hydrogen bonds with residues from the tripod to anchor the VP9 trimer onto the VP4 trimer (Fig. 6b). A density map of the VP9 trimer at an overall resolution of 3.8 Å was obtained by iterative 3D classifications and subpopulation averaging of the VP9 density at multiple positions, allowing us to resolve an atomic model for almost full-length VP9 (covering residues 4–283) (Fig. 6c). The atomic model of the VP9 protomer is largely identical to the previously reported crystal structure of VP9 (PDB ID: 1W9Z, missing ~30 N-terminal residues of the stalk region, RMSD of 0.84 Å for 257 Cα atoms). The VP9 stalk adopts a helix–loop–helix–loop–helix conformation, and multiple residues are involved in interprotomer interactions (Fig. 6c). The extended loops connecting helices α1 and α2 serve as the internal hinge for tilting of the *C*-terminal parts of VP9 trimers.

To study possible molecular events after BAV is internalized in the endosome, we examined the conformational changes in BAV surface proteins triggered by low pH treatment. A solution containing BAV particles turned turbid after treatment with a buffer at pH 5.0. Analysis of protein composition by SDS-PAGE revealed that VP9 was released from the virions as a soluble component, while VP4 remained in the virions (Fig. 6d), suggesting that acidic treatment induced the

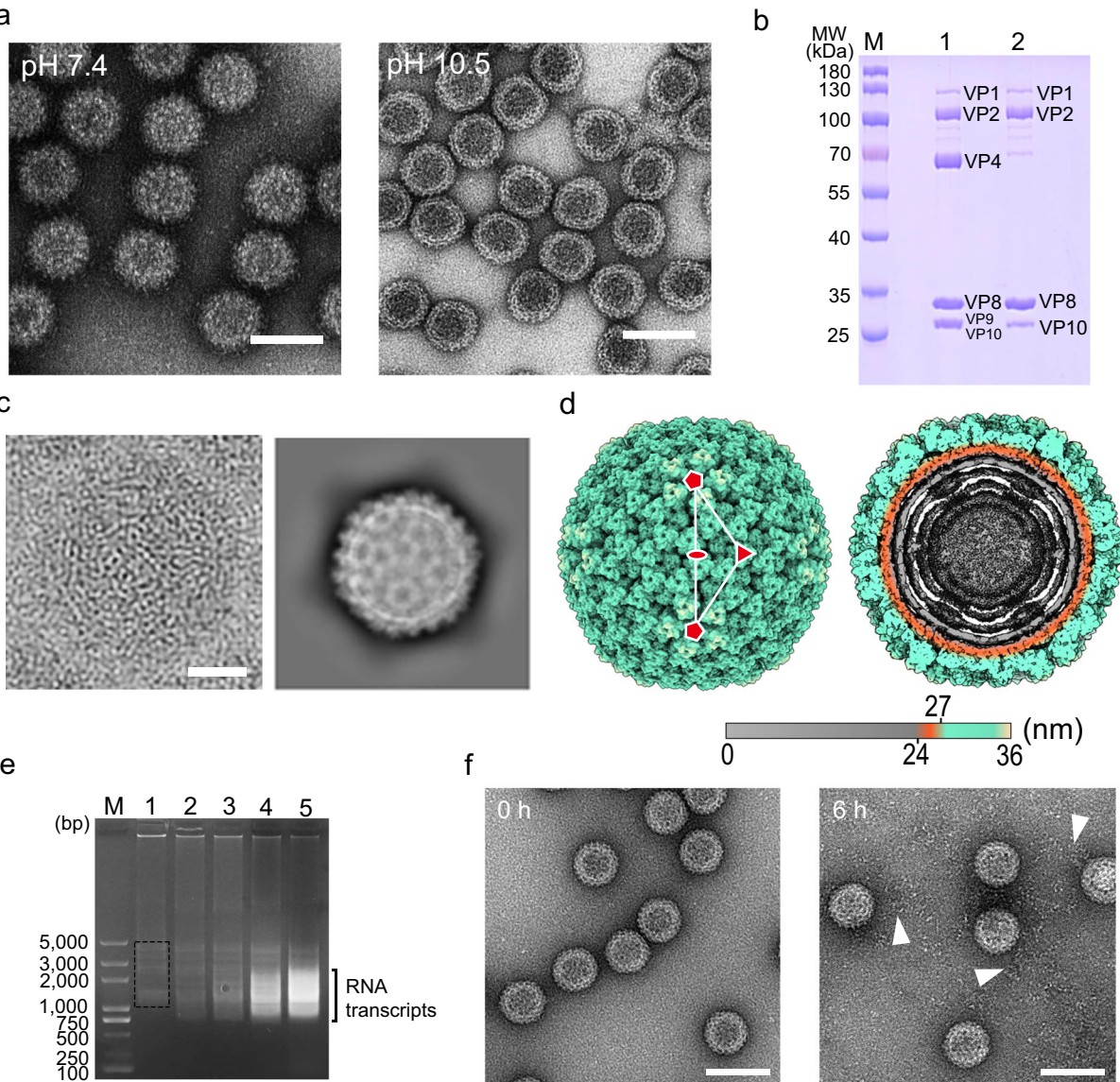

**Fig. 5 | Alkaline treatment and core particle structure. a** Negative stain TEM of BAV particles (left) and alkaline buffer-treated BAV particles (right). Scale bar: 100 nm. **b** Electrophoretic analysis of BAV proteins using SDS-PAGE. Lane 1, purified BAV particles. Lane 2, alkaline buffer-treated BAV particles (cores). Note that VP9 and VP10 show similar migration patterns in SDS-PAGE, but based on structural analysis, only VP9 is absent from cores. **c** 2D image (left) and class averaging (right) of core particles. Scale bar: 20 nm. **d** Surface representation (left) and cutaway view (right) of core particle. **e** Agarose gel electrophoresis analysis of transcriptional reaction mixtures. BAV cores were incubated in transcription buffer at 30 °C for 0 (Lane 1), 0.5 (Lane 2), 1 (Lane 3), 3 (Lane 4), and 6 h (Lane 5). dsRNA genomic segments are indicated in the dashed box, and RNA transcripts appear as smeared bands. dsDNA markers are shown in Lane M. **f** Negative-stain TEM of core particles in transcription buffer at 0 (left) and 6 h (right). Fiber-like materials (RNA transcripts) indicated by white arrowheads are visible around transcribing BAV core particles. Scale bar: 100 nm.

dissociation of VP9 from VP4. In the cryo-EM structure at overall 6 Å-resolution based on ~1400 acidified virus particles (Supplementary Fig. 10), 16-nm rod-shaped projections (the density of VP4 at ~15 Å local resolution) arise only from VP10A at the type II channel (Fig. 6e), suggesting that the specific shedding of spikes that occurs on partial particles may be required to form this acidified conformation.

## Discussion

Based on the structural similarity of RdRp (VP1) and VP4, BAV is probably more closely phylogenetically related to orbivirus BTV than to rotavirus RRV; however, there is no counterpart to the cementing protein VP10 in BTV virions, and the composition and distribution of surface proteins in the two viruses is also quite different. The triple-layered BTV virion contains an inner layer consisting of 120 VP3 monomers, a middle layer of 260 VP7 trimers, and an outer layer formed by 60 VP2 trimers (for receptor binding) and 120 VP5 trimers (for membrane penetration)[16]. VP5 trimers are located at the type II and type III channels, while each VP2 trimer sits on four VP7 trimers. This arrangement of BTV surface proteins enables the penetration protein of BTV to refold even when the receptor binding protein remains on the virions, upon exposure to low pH conditions[11]. In contrast, in the double-layered full BAV particles, 120 spikes made up of a VP4 trimer (for membrane penetration) and VP9 (for receptor binding) emanate from VP10 located at both type II and type III channels. The binding affinity of spikes at the two channels appears different, resulting in detachment of half the spikes when the partial particles form. In addition, if VP4 functions as inferred from paralogs, VP9 should detach from the pedestal domain to allow conformational changes in the unfurling domain of VP4. Indeed, on acidified particles, the rod-shaped projections of VP4 are reminiscent of the six-helix stalks formed by BTV VP5[11].

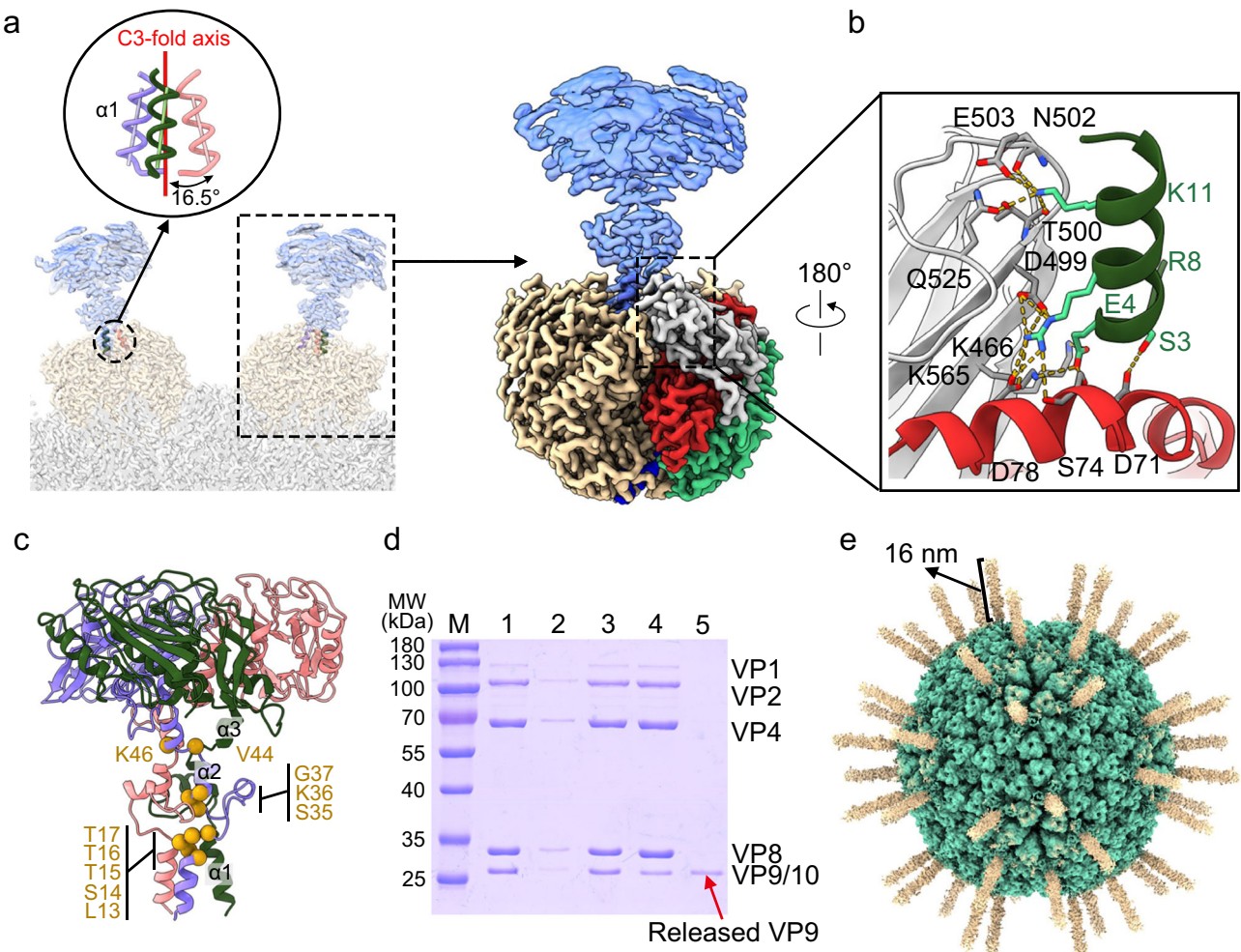

**Fig. 6 | VP9. a** (Left panel) *N*-terminal α1 helices of the VP9 trimer (in transparent blue, based on our C5 reconstruction) forming a tripod to insert into the central cavity of the VP4 trimer (in transparent beige). (Right panel) Density map view of one spike, with one VP4 subunit being color-coded by domain (similar to Fig. 4d). **b** Zoomed-in view of the black box from a, showing interactions of helix α1 from one VP9 subunit with the pedestal domain (gray) and the unfurling domain (red) from VP4. **c** Atomic model of VP9 based on local reconstruction of VP9 (Supplementary Fig. 4). The three VP9 protomers (colored lavender, dark green and pink) are shown in the ribbon diagram, highlighting the *N*-terminal stalk region of VP9 (made up mainly of α1–α3). The residues involved in interprotomer interactions within the stalk region are labeled and denoted by golden spheres in one VP9 molecule (in lavender). **d** SDS-PAGE analysis of the protein composition of purified BAV particles (in PBS, pH 7.4, Lanes 1–3) and BAV particles in acidic buffer (citrate–phosphate buffer, pH 5.0, Lanes 4–5). Lane 1, concentrated BAV particles; Lane 2, pelleted BAV particles after centrifugation at 10,000 × *g*, showing only a small amount of particles are precipitated; Lane 3, supernatant after centrifugation at 10,000 × *g*, showing most BAV particles are intact in solution; Lane 4, pelleted particles after centrifugation, showing most particles are precipitated; Lane 5, supernatant after centrifugation, showing VP9 protein has been released into solution. **e** Cryo-EM structure of BAV particles after acid treatment (pH 5.0), showing VP4 density as rod-shaped projections (colored in beige) on the viral core.

The flexibility of VP9 on the virus surface is reminiscent of trimeric spikes from many enveloped viruses, such as HIV-1 and SARS-CoV-2. Env proteins of native HIV-1 virions adopt different tilt angles with respect to the virus membrane, which is mediated by a flexible gp41 stalk[30]. For SARS-CoV-2, spike (S) protein trimers are not all perpendicular to the virion surface, and the membrane-proximal stalk region connecting the S ectodomain and the transmembrane domain functions as a flexible hinge that allows the ectodomain to tilt in variable directions[31]. Considering the large number of VP9 trimers on the BAV particles, the structural plasticity of loops in the stalk region of VP9 may facilitate the interactions of rigid head domains with multiple receptor molecules on host cells with variable binding orientations.

The sedoreovirus core particles are molecular machines for synthesis, capping, and export of viral mRNA transcripts, and thus have been structurally studied extensively for a long time[2,3]. Rotavirus core particles can be purified from infected cells[32], whereas BTV

particles have been treated with a combination of chymotrypsin and sodium N-lauroyl sarcosinate to prepare their cores[33]. Although VP10 is the smallest structural protein in BAV virions, its function as a cementing protein makes it a critical player in the virus assembly and dynamic transitions. Because the electrostatic attraction between VP10 and VP4 is weakened at high pH, BAV core particles can be conveniently obtained by alkaline treatment of BAV particles. Recently, cryo-EM studies of mud crab reovirus, a putative member of the family *Sedoreoviridae*, revealed that a protein component named VP11, with unknown function, is located at type II and III channels in a DLP structure[34], indicating that there may be VP10-like cementing proteins in other sedoreoviruses.

Based on our cryo-EM structures of four types of BAV particle (full, partial, acidified, and core) and in vitro biochemical data, we propose a sequence of four basic molecular events for delivery of BAV cores into the cytoplasm of an infected cell: Step 1. The viral infection is initiated

by VP9, which recognizes and attaches to unknown cellular receptors in a multivalent-binding manner. Step 2. After BAV particles are taken up via endocytosis, endosomal acidification induces removal of VP9 to release the lock on VP4. Step 3. VP4 undergoes a substantial conformational rearrangement from the "pre-penetration" form to the "post-penetration" form, perforating and rupturing the endosomal membrane, while its anchoring domain remains attached to the underlying VP10 molecule. Step 4. VP4 dissociates from the outer capsid shell and cores are then released into the cytoplasm of the infected cell for BAV replication. BAV entry into the cell is a highly ordered process with stepwise detachment of viral spikes, as reflected by the variation in spike arrangement on full particles (120 spikes containing VP4 and VP9), partial particles (60 spikes containing VP4 and VP9), acidified particles (60 spikes containing VP4), and core particles (no spikes). Although alkaline treatment and proteolytic cleavage facilitate the production of cores, the molecular mechanism underlying in vivo removal of spikes from the virions remains to be investigated. More studies of the dynamics of BAV structural proteins would not only provide a new reovirus model for evaluating the entry mechanism of non-enveloped viruses, but also help to identify potential targets for antiviral therapies against lethal pathogens in the genus *Seadornavirus*.

## Methods

### Growth characteristics of BAV strain YN15-126-01 in cell culture

Banna virus (BAV) strain YN15-126-01 was isolated from the *Culex tritaeniorhynchus* mosquito collected in 2015 in Yunnan, China[24]. C6/36 cells were maintained at 28 °C under 5% $CO_2$ in RPMI medium (Gibco) supplemented with 10% fetal bovine serum (FBS; Gibco) and 1% penicillin/streptomycin. BHK-21, PK15, Vero, A549, and Huh-7 cells were grown at 37 °C under 5% $CO_2$ in Dulbecco's minimal essential medium (Gibco) supplemented with 10% FBS and 1% penicillin/streptomycin, while LMH cells were cultured in Dulbecco's Modified Eagle's Medium/Nutrient Mixture F-12 (Gibco) with 10% FBS and 1% penicillin/streptomycin at 37 °C under 5% $CO_2$. C6/36, LMH, BHK-21, PK15, Vero, A549, and Huh-7 cells grown in six-well plates were infected with virus at R/C = 1 or 1000. After inoculation, the CPEs associated with infection were monitored; 200 μl of cell supernatant was collected every day (from day 0 to day 7) and then replenished with 200 μl of fresh medium.

Viral RNA was extracted from the cell culture supernatant using a QIAamp Viral RNA Mini Kit (QIAGEN) according to the manufacturer's instructions. A viral RNA standard (125-nt) was produced by using an in vitro T7 High Yield RNA Transcription Kit (Vazyme) and serial dilutions (10-fold) of the standard RNA were used to generate a standard curve for RT-qPCR. RT-qPCR for the detection of viral RNA was performed using a One Step TB Green PrimeScript PLUS RT-PCR Kit (TaKaRa) according to the manufacturer's recommendations with a thermocycler (Bio-Rad CFX96 Real-Time System). The primers for SYBR Green RT-qPCR targeted VP11 of BAV: BAV-F (5′–TACAACCTCAC GGTGTGACAAGG – 3′) and BAV-R (5′–TCATCAACGGCCAGTGG CGG – 3′). The viral RNA copy number (copies/ml) in cell culture was calculated based on the standard curve [$y = -3.807x + 52.23$, $R^2 = 0.9991$, where y is the Ct (cycle threshold) value, and x is the log of the RNA copy number]. The experiment was repeated three times.

### BAV propagation and purification

BAV was propagated in C6/36 cell cultures at 28 °C for 6 days. To recover mature BAV particles, the supernatant was filtered through 0.22 μm membranes (Millipore), and centrifuged at $140,000 \times g$ for 2 h at 4 °C with a 20% (w/v) sucrose cushion. The pelleted virions were resuspended in 1× phosphate-buffered saline (PBS; pH 7.4), and were then subjected to further sucrose gradient (10%–80%) purification at $171,500 \times g$ for 3 h at 4 °C. BAV virions collected from gradients were

concentrated to ~4 mg/ml in 1× PBS, measured by using a spectrophotometer (NanoPhotometer). Core particles were prepared by a similar procedure, except the pelleted virions were resuspended in an alkaline buffer (200 mM NaCl, 50 mM $Na_2HPO_4$, adjusted to pH 10.5 with NaOH). The final concentration of purified cores was ~4 mg/ml (also in 1× PBS). Freshly purified particles were examined by negative-stain transmission electron microscopy (TEM). Three microliters of virion sample were applied onto a glow-discharged carbon-film coated grid on ice for 20 s. After removal of excess solution, the grid was negatively stained with 2% (w/v) uranyl acetate. The grids were imaged using a Talos L120C (Thermo Fisher Scientific) equipped with a CETA 16 M detector.

### Cryo-EM grid preparation and data collection

For cryo-EM, 2.5 μl of purified virion sample was applied onto a glow-discharged holy-carbon grid (QuantiFoil, R1.2/1.3). The grid was plunge-frozen in liquid ethane using a Vitrobot Mark IV plunger (ThermoFisher Scientific) after blotting for 3 s under 100% humidity at 16 °C. The vitrified grids were loaded into a CRYO ARM 300 electron microscope (JEOL) equipped with a K3 direct electron detector (Gatan). Movies were recorded using SerialEM[35] at a nominal magnification of 50,000× using a defocus range of 0.5–2.5 μm in the super-resolution mode, corresponding to a pixel size of 0.475 Å. Each movie stack was dose-fractionated to 40 frames with a total electron exposure of ~40 e⁻/ Å².

### Classification of BAV particles

Frames in each movie were 2× binned (giving a pixel size of 0.95 Å) and motion-corrected using MotionCorr2 implemented in RELION[36]. After manual selection, 5943 dose-weighted micrographs for BAV virions and 2163 micrographs for high pH-treated virions (cores) were imported into cryoSPARC[37], respectively, followed by contrast-transfer function estimation (Supplementary Fig. 3). Virus particles were manually picked and subjected to reference-free two-dimensional (2D) classification using cryoSPARC. Representative classes were used as references for template-based particle picking; 111,846 particles were automatically picked for BAV virions, and 49,730 particles for cores, respectively. Best particles were then selected by the "2D classification" and "Select 2D Classes" jobs in cryoSPARC. Ab initio reconstruction and heterogeneous refinement with icosahedral symmetry imposed revealed that there were two typical 3D classes for BAV virions (i.e., full and partial particles) and mainly one 3D class for cores (38,945 particles), respectively. Because subsequent data processing for the three types of particles was performed by a similar strategy, only the detailed procedure for full particles is described below.

### Local reconstruction of fivefold vertex and asymmetric reconstruction of RdRp

Full particles selected in cryoSPARC were re-extracted in RELION (3× binned, giving a pixel size of 2.85 Å). Icosahedral reconstruction with I3 symmetry imposed was performed in RELION to generate a 5.73 Å resolution map, in which two C5-fold vertices are aligned along the Z-axis. The refined STAR file was expanded with I3 symmetry using the 'relion_particle_symmetry_expand' command to generate a new STAR file, which contained 60 orientations for each particle and five orientations relative to each vertex. To obtain the subparticle reconstruction of the C5-fold vertex, the redundant four entries for each vertex were removed by using the Linux command 'sort', as described previously in ref. 38. Then we extracted the particles (unbinned, giving a pixel size of 0.95 Å) using the "Particles extraction" tool with the sorted STAR file as the input and set the "recenter on – X Y Z (pixel)" parameters as 0, 0, 95 (the new center was ~270 Å away from the centroid of the icosahedral map along the fivefold Z-axis). The extracted subparticles at fivefold vertices were reconstructed by the

'relion_reconstruct' command with C5 symmetry imposed. The resulting reconstruction served as the reference for further 3D refinement to generate a map at 2.6 Å resolution. The resolution was estimated on the basis of the "gold standard" Fourier Shell Correlation = 0.143 criterion[39].

To solve density for VP1 (RdRp) underneath the C5 vertex, we conducted a C1 (asymmetric) 3D classification focused on the RdRp region in the 2.6 Å map. The particle STAR file from the last 3D refinement job was expanded with C5 symmetry. A pancake-shaped mask encompassing RdRp density was created and finely adjusted to exclude capsid volume as much as possible using ChimeraX[40]. Asymmetric 3D classification of the masked region was then performed using symmetry-expanded particles without alignment. The classification yielded 12 classes, of which eight classes displayed clear RdRp density in five different positions. The particles from classes representing a single position were selected and duplicative particles were removed by the 'sort' command based on _rlnMaxValueProbDistribution. The subparticles were reconstructed by the 'relion_reconstruct' command with no symmetry imposed, and further 3D refinement yielded a reconstruction of the C5 vertex (containing one RdRp) at overall 3.0 Å resolution (Supplementary Fig. 3).

### Local refinement of VP9

The local resolution of the main body of VP9 in the 2.6 Å map only reached to ~5 Å, not sufficient for model building. To improve the resolution of VP9, we created a spherical mask encompassing one spike (VP9 plus VP4) near the C5-fold axis using ChimeraX (Supplementary Fig. 4). Three-dimensional classification of the masked region without alignment was performed using C5-symmetry expanded particles. Particles from one class showing the best density for the VP9 trimer were selected. The "Particle subtraction" job in RELION was then performed with "Do center subtracted images on mask", and the resulting subparticles were imported into cryoSPARC for 3D refinement. Two rounds of ab initio reconstruction and heterogeneous refinement generated four 3D classes, one of which exhibited the correct morphology of spike with C3 symmetry. Non-uniform refinement was performed for the particles from this class. Local refinement of the density for VP9 and part of VP4 was then performed, and generated a map at an overall 3.8 Å resolution.

### Cryo-EM model building and analysis

The starting atomic models for BAV structural proteins (except VP9) were predicted by AlphaFold2[41] on a local server, based on the sequences of strain YN15-126-01. Molecular docking of the AlphaFold2 models into the refined cryo-EM maps postprocessed by DeepEMhancer[42] revealed that VP1 and VP8 showed reasonable fitting, while other models were fitted with poor quality, especially for VP4 and VP10, which we had to manually rebuild. VP9 crystal structure (PDB ID: 1W9Z) was docked well into the local reconstruction map. The models were improved by iterative cycles of real-space refinement and interactive optimization using Phenix[43] and Coot[44]. Structural validation was performed using Molprobity[45]. The cryo-EM data collection and refinement statistics are summarized in Supplementary Table 1. The radial density profiles of virions were calculated using the program e2proc3d.py[46]. Protein secondary elements were assigned by DSSP[47]. Hydrogen bonds and salt bridges were identified using PDBePISA[48]. Figures were prepared using ChimeraX.

### Cell-free transcription reaction

Purified core particles were incubated in transcription buffer [4 mM rATP, 2 mM rUTP, 2 mM rCTP, 2 mM rGTP, 1 mM S-adenosyl-L-methionine (SAM), 5 mM DTT, 10 mM $MgCl_2$, 100 mM Tris-HCl (pH 7.5), and 2 U/µl recombinant RNase inhibitor (RRI, Takara no. 2313 A)] for 0, 0.5, 1, 3, or 6 h at 30 °C. The reaction was terminated by adding EDTA to the mixture to 50 mM, and the reaction products

were electrophoresed on agarose gels that were stained with Goldview (Biosharp No. BS357A). Negatively stained micrographs of the 6 h reaction mixture were also used to evaluate RNA transcription.

### Treatment of BAV particles by acidic buffer

For the acidic treatment, BAV suspension in 1× PBS was diluted tenfold with citrate–phosphate buffer (50 mM citric acid, 100 mM $Na_2HPO_4$, pH 5.0), and incubated on ice for 30 min. The mixture was then centrifuged at 10,000 × g for 10 min; the pellet and the supernatant were separated and boiled in SDS-PAGE loading buffer for 10 min before being examined by SDS-PAGE. To prepare the grid for cryo-EM, the virus sample was first deposited onto a Lacey-carbon grid (Ted Pella, #01824) for 30 s. After removal of excess virus solution, buffer (pH 5.0) was applied onto the grid for 2 min. The grid was then loaded onto the Vitrobot Mark IV plunger for blotting for 4.5 s under 90% humidity at 16 °C before being plunge-frozen in liquid ethane.

### Statistics and reproducibility

TEM and SDS-PAGE images shown in figures are representative of at least three independent experiments.

### Reporting summary

Further information on research design is available in the Nature Portfolio Reporting Summary linked to this article.

## Data availability

The full genome sequences of BAV YN15-126-01 were submitted to GenBank, and the accession numbers for segments 1 to 12 are OR004518, OR004519, OR004520, OR004521, OR004522, OR004523, OR004524, OR004525, OR004526, OR004527, OR004528, OR004529. Cryo-EM density maps have been deposited in the Electron Microscopy Data Bank under accession codes EMD-37378 (icosahedral full particles), EMD-37379 (icosahedral partial particles), EMD-37380 (icosahedral core particles), EMD-36870 (C5 vertex of full particles), EMD-36880 (C5 vertex of partial particles), EMD-36881 (C5 vertex of core particles), EMD-36871 (C1 vertex of full particle showing VP1), and EMD-36872 (local reconstruction of VP9). The corresponding atomic models have been deposited in the Protein Data Bank with accession codes 8W9P, 8W9Q, 8W9R, 8K42, 8K49, 8K4A, 8K43, and 8K44, respectively. Previous reported atomic model of BTV VP5 can be accessed under accession code 3J9E.

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

## Acknowledgements
This work was supported by the Strategic Priority Research Program of the Chinese Academy of Sciences (XDB0490401 to S.C.), by the National Key Research and Development Program of China (2022YFC2303300 to S.C.), and by the Youth Innovation Promotion Association CAS (2022341 to G.R.). We thank Prof. Peng Gong for informative discussion about viral RdRps. We also thank the Center for Instrumental Analysis and Metrology of Wuhan Institute of Virology for technical assistance.

## Author contributions
S.C. and Z.Y. conceived the project. Z.L. and H.X. prepared virus samples. G.R. collected EM data. S.C. and Z.L. processed the data. Y.F., T.C., and K.T. helped with virus preparation under various conditions. S.C. and Z.L. interpreted results and wrote the paper with contributions from all the other authors.

## Competing interests
The authors declare no competing interests.
