## [Peer Review File · Nature Communications]

Cryo-EM structures of Banna virus in multiple states reveal stepwise detachment of viral spikesReviewers' Comments:

Reviewer #1:

Remarks to the Author:

In this manuscript, Li et al. have used a high-resolution cryo-EM technique to provide the most comprehensive atomic-level structural description of the Banna virus, which belongs to the non-turreted subgroup in the Reoviridae similar to rotavirus and bluetongue virus (BTV). The authors describe the structure of each of the 6 components that form the double-layered capsid structure based on underlying icosahedral symmetry. The structures of all the components of the virion, VP1 the viral polymerase residing inside; VP2, which forms the inner capsid with T=1 symmetry; VP8, which forms the outer capsid layer with T=13 symmetry and the associated unique cement protein VP10; and VP9 and VP4 which constitute the distal spikes. Although there are a lot of similarities between BTV and, to some extent, rotavirus, there are quite a few fascinating differences that this study brings out very nicely. In addition to mature particles, the authors also determined the structure of the partial and core structures at high resolution. Based on these structures and in vitro biochemical data, the authors suggest possible stages of how the cores are delivered into host cells.

This well-written paper provides a complete atomic-level description of the complex architecture of the Banna virus, highlighting the contrasting features with other non-turreted reoviruses. The high-resolution cryoEM is excellently done using the appropriate image processing procedures for relaxing and imposing the icosahedral symmetry. Methodological details are adequately described, and the figures are appropriately designed to describe the complex capsid organization.

Minor comments and suggestions.

It is a bit puzzling how the production of the core structure, either by high pH or the chymotrypsin treatment, fits in with the stage of endogenous transcription during virus infection following endosomal virus entry. Have the authors checked if the cores obtained from either of these treatments are transcriptionally active?

Figure 1: VP9 is colored in different shades of blue in panel h? – please, correct.

Methods: Lines 420-428 – make clear which software was used here. Was it all cryo-SPARC?

References:

Line 57: along with ref 8, include Hill et al. (DOI: 10.1038/9347), the first paper to coin the phrase turreted and smooth.

Lines 167-168: include references on in situ structures of RdRP in rotavirus Jenni et al., DOI: 10.1016/j.jmb.2019.06.016 .

Reviewer #2:

Remarks to the Author:

The authors present a comprehensive structural analysis of infectious Banna viruses using single particle electron cryomicroscopy. Banna virus, with reported infections of humans, is a prototype of the Seadornavirus genus of viruses, and its structure presented here is important for understanding its architecture and organization of functional elements (viral polymerase, core particle, membrane penetration spike proteins). The manuscript reads well with a clear presentation of the structural results. The interpretation of the molecular structures is appropriate, and the authors provide sufficient methodological details. The figures are excellent.

Like other non-enveloped reoviruses, Banna virus delivers a transcriptionally active core particle into the cytoplasm of the host cells. For this, a membrane disruption step must occur that is catalyzed by viral outer layer or spike proteins. Among different reoviruses, these membrane penetration proteins

(VP4 and VP9 in case of Banna virus) have a similar function, but they have vastly different molecular structures. To visualize the inferred pH-induced conformational change of the spike proteins, I suggest that the authors explore a more detailed structural analysis of low-pH-treated virions. Even a low-resolution reconstruction from a modest number of such virions would be valuable in order to understand the proposed conformational changes of VP4 and VP9. The authors treated virions at pH 3.4, which is quite extreme, and a pH of 4.5-5.0, as physiological in endosomes, might yield less aggregation and a better sample for electron cryomicroscopy. Or perhaps virions can be first deposited on thin-carbon layer grids before lowering the pH. I believe that if a more detailed structural analysis of pH-treated virions was included in this manuscript, the study would certainly be an important step forward in our understanding the biology of human reoviruses and its significance would not be limited to specialists in the reovirus field only.

Comments, questions, and suggestions:

Maybe there is a better title for the manuscript. The "stepwise detachment of viral spikes" has no physiological role in this context (e.g. preparation of core particle by alkaline treatment).

What is the consensus in the field for annotation of the Banna virus VP4 protein? In UniProt, VP4 is annotated as "outer capsid protein" (<https://www.uniprot.org/uniprotkb/Q9INI3/entry>). Here, the authors classify it as spike protein. In case it is an outer capsid protein, as UniProt suggests, shouldn't the full virion be called triple-layered particle instead of double-layered particle? Or is VP4 considered part of the VP8/VP10 layer? I agree with the authors that it should rather be assigned as spike protein.

Why do VP10A and VP10B specifically intercalate between the VP8 trimers Q and R? How does this cleft structurally differ from the other interfaces between VP8 trimers around the pseudo six-folds of the channels II and II?

How similar are the channels II and III, e.g. if they are superimposed?

Spike occupancy on channel III appears to be "all or nothing" as there are particles with either 60 or 120 spikes. Do the authors have an explanation for this? Why are there no particles with partial spike occupancies? Is there density at low contour levels for partially occupied spike positions?

In the asymmetric RdRp reconstruction, are there any additional structural changes in the VP2 proteins (in addition to the N terminus) induced by RdRp binding, for instance around the five-fold pore in the apical domain of VP2A?

Line 233: "However, the corresponding loop region in BAV VP4 (Q319–K330) is not 234 likely to form a similar structural motif because of the presence of two destabilizing glycine 235 residues (G321 and G329) in this region (QYGNTSASEFGK) (Fig. 4e)." I'm not convinced of this argument, given that the VP5 from BTV and VP4 from BAV are paralogs, I would not be surprised if the same loop in BAV has a similar structure and function if rearranged.

Fig. 6d: The description of the conditions for lines A, B, C, and D are confusing. Were A and C treated identical (except that C was low pH-treated)? The caption says that A is "concentrated" and C is "pelleted". The same fractions for the neutral and low-pH treated virus should be analyzed by SDS-PAGE.

Minor comments:

Abstract: The authors say that VP10 "plays pivotal roles in the assembly of BAV virions", can they explain what roles? Also, why is it relevant to mention here that it has a high pI value?

Line 33: conformational changes in the membrane penetration protein.

Line 93: "unique structural features", please specify.

Line 99: nucleotide similarity (>95%), do the authors mean identity? There are similar amino acids, e.g. conservative substitutions, but what were similar bases?

Line 101: the host cells were inoculated with virus, not the virus. Change to "host cells were inoculated with virus".

Line 102: How is the ratio viral RNA/cells (R/C) calculated? Is it meant number of virions to number of cells? Or how is viral RNA determined? Please clarify.

Line 118, 120: not an electron density map, Coulomb potential or just density map.

Line 131: Why is it "pseudo T = 2" and not just "T = 2" lattice?

Line 165: DALI server, please add citation.

Line 168: Citation, S. Jenni, et al., In situ Structure of Rotavirus VP1 RNA-Dependent RNA Polymerase, *Journal of Molecular Biology*, <https://doi.org/10.1016/j.jmb.2019.06.016> could be added here, as not all N-terminal VP2 tails interacting with the rotavirus RdRp were modeled in reference 25.

Line 169: Delete "Unexpectedly".

Line 299: "more phylogenetically related to orbivirus BTV" more than which other virus?

Line 312: replace "correctly", with "as inferred from paralogs".

Line 384: what are Ct values?

Line 648: Please remove the statement that the map was contoured at 1.6 sigma. The value of 1.6 is meaningless, as it depends on the size of the box of the reconstruction and the masking of the density. Meaningful would be at what absolute value of Coulomb potential the map was contoured, or at what absolute density level (e.g. protons per cubic Angstrom).

Line 650: Please define ASU (asymmetric unit).

Line 659: doesn't look like map was filtered at 6 angstrom resolution, looks like one can see individual atoms.

Line 688: five types of VP8 trimers

Line 690: three types of channels

Lines 751 and 752: color description of domains, needs editing?

Fig. S1b: What is dpi, please define?

Caption Fig. S1b: What is CPE, please define?

Fig. S8c: The structure of the Banna virus VP4 protomer should be added next to the BTV VP5 protomer in the same representation for direct comparison.

Caption Fig. 5a: analysis of alkaline-treated core particles, is this after purification of core particles, e.g. by pelleting?

Caption Fig. S9, line 138: Replace "Dynamics" with "Structures"

Fig. S8e. Why are the VP2 and VP8 bands much fainter in lane C (virions incubated with 400 mM MgCl₂ in the absence of trypsin) compared to the control samples in lane A or B?

In response to Referee #1:

Reviewer #1 (Remarks to the Author):

In this manuscript, Li et al. have used a high-resolution cryo-EM technique to provide the most comprehensive atomic-level structural description of the Banna virus, which belongs to the non-turreted subgroup in the Reoviridae similar to rotavirus and bluetongue virus (BTV). The authors describe the structure of each of the 6 components that form the double-layered capsid structure based on underlying icosahedral symmetry. The structures of all the components of the virion, VP1 the viral polymerase residing inside; VP2, which forms the inner capsid with T=1 symmetry; VP8, which forms the outer capsid layer with T=13 symmetry and the associated unique cement protein VP10; and VP9 and VP4 which constitute the distal spikes. Although there are a lot of similarities between BTV and, to some extent, rotavirus, there are quite a few fascinating differences that this study brings out very nicely. In addition to mature particles, the authors also determined the structure of the partial and core structures at high resolution. Based on these structures and in vitro biochemical data, the authors suggest possible stages of how the cores are delivered into host cells.

This well-written paper provides a complete atomic-level description of the complex architecture of the Banna virus, highlighting the contrasting features with other non-turreted reoviruses. The high-resolution cryoEM is excellently done using the appropriate image processing procedures for relaxing and imposing the icosahedral symmetry. Methodological details are adequately described, and the figures are appropriately designed to describe the complex capsid organization.

Response: Thank you for these comments.

Minor comments and suggestions.

It is a bit puzzling how the production of the core structure, either by high pH or the chymotrypsin treatment, fits in with the stage of endogenous transcription during virus infection following endosomal virus entry. Have the authors checked if the cores obtained from either of these treatments are transcriptionally active?

Response: Thank you for pointing out the activity issue with the viral core. Structurally speaking, following the definition by Mohd Jaafar *et al.* (DOI: 10.1099/vir.0.80578-0), the core is the remaining viral particle after removal of VP4 and VP9 (as we mention in lines 84–85). We agree that the core should be functional in this state. We thus examined transcriptional activity and found that the core that was derived from the alkaline treatment had the expected activity, while the “core” obtained by the digestion method lacked consistent activity, possible because of “over-digestion”. To reflect the importance of core activity, in the revised manuscript, we have removed the paragraph about chymotrypsin/trypsin digestion and the related content in the Methods section. We have added the following sentences in the Results section (lines 254–260): “To test the transcriptional activity of the core, cell-free transcription reaction was performed in similar conditions to those previously described, and the reaction products were examined using

agarose gels on which newly synthesized RNA transcripts were shown to be accumulated in a time-dependent manner (Fig. 5e). Negative-staining TEM showed that there were separately distributed “fibers” surrounding the incubated core particles (Fig. 5f, right panel), also indicating that the alkaline-treated virus particles were transcriptionally active.” Meanwhile, we added some sentences to the Methods section (lines 498–505): “Cell-free transcription reaction. Purified core particles were incubated in transcription buffer [4 mM rATP, 2 mM rUTP, 2 mM rCTP, 2 mM rGTP, 1 mM S-adenosyl-L-methionine (SAM), 5 mM DTT, 10 mM MgCl₂, 100 mM Tris-HCl (pH 7.5), and 2 U/μl recombinant RNase inhibitor (RRI, Takara no. 2313A)] for 0, 0.5, 1, 3, or 6 h at 30 °C. The reaction was terminated by adding EDTA to the mixture to 50 mM, and the reaction products were electrophoresed on agarose gels that were stained with Goldview (Biosharp No. BS357A). Negatively stained micrographs of the 6-h reaction mixture were also used to evaluate RNA transcription.”

We have added new Fig. 5e and Fig. 5f to present the core activity, and moved the original Fig. 5e to be Supplementary Fig. 9a.

Caption:

e, Agarose gel electrophoresis analysis of transcriptional reaction mixtures. BAV cores were incubated in transcription buffer at 30 °C for 0 (Lane 1), 0.5 (Lane 2), 1 (Lane 3), 3 (Lane 4), and 6 h (Lane 5). dsRNA genomic segments are indicated in the dashed box, and RNA transcripts appear as smeared bands. dsDNA markers are shown in Lane M.

f, Negative-stain TEM of core particles in transcription buffer at 0 (left) and 6 h (right). Fiber-like materials (RNA transcripts) indicated by white arrowheads are visible around transcribing BAV core particles. Scale bar: 100 nm.

Figure 1: VP9 is colored in different shades of blue in panel h? – please, correct.

Response: VP9 has weaker density in the icosahedral reconstruction of the full viral particle than the other capsid components, and thus we used a transparent representation of VP9 in Fig. 1g. In Fig. 1h, we used a similar approach to coloring VP9 molecules. In the revised manuscript, the central model in Fig. 1h is colored normally without applying transparency to represent VP9 density.

Methods: Lines 420-428 – make clear which software was used here. Was it all cryo-SPARC?

Response: Yes, it was all performed with cryoSPARC. To clarify this, we have modified some sentences (lines 425–429): “... reference-free two-dimensional (2D) classification using cryoSPARC” ... “the “2D classification” and “Select 2D Classes” jobs in

cryoSPARC.”

References:

Line 57: along with ref 8, include Hill et al. (DOI: 10.1038/9347), the first paper to coin the phrase turreted and smooth.

Lines 167-168: include references on in situ structures of RdRP in rotavirus Jenni et al., DOI: 10.1016/j.jmb.2019.06.016 .

Response: Modified as requested, thank you.

In response to Referee #2:

Reviewer #2 (Remarks to the Author):

The authors present a comprehensive structural analysis of infectious Banna viruses using single particle electron cryomicroscopy. Banna virus, with reported infections of humans, is a prototype of the Seadornavirus genus of viruses, and its structure presented here is important for understanding its architecture and organization of functional elements (viral polymerase, core particle, membrane penetration spike proteins). The manuscript reads well with a clear presentation of the structural results. The interpretation of the molecular structures is appropriate, and the authors provide sufficient methodological details. The figures are excellent.

Response: Thank you for these comments.

Like other non-enveloped reoviruses, Banna virus delivers a transcriptionally active core particle into the cytoplasm of the host cells. For this, a membrane disruption step must occur that is catalyzed by viral outer layer or spike proteins. Among different reoviruses, these membrane penetration proteins (VP4 and VP9 in case of Banna virus) have a similar function, but they have vastly different molecular structures. To visualize the inferred pH-induced conformational change of the spike proteins, I suggest that the authors explore a more detailed structural analysis of low-pH-treated virions. Even a low-resolution reconstruction from a modest number of such virions would be valuable in order to understand the proposed conformational changes of VP4 and VP9. The authors treated virions at pH 3.4, which is quite extreme, and a pH of 4.5-5.0, as physiological in endosomes, might yield less aggregation and a better sample for electron cryomicroscopy. Or perhaps virions can be first deposited on thin-carbon layer grids before lowering the pH. I believe that if a more detailed structural analysis of pH-treated virions was included in this manuscript, the study would certainly be an important step forward in our understanding the biology of human reoviruses and its significance would not be limited to specialists in the reovirus field only.

Response: Thank you for raising this important issue. We carried out the experiment as suggested, and new data have been added to the Results section (lines 286–291): “In the cryo-EM structure at overall 6-Å-resolution based on ~1,400 acidified virus particles (Supplementary Fig. 10), 16-nm rod-shaped projections (the density of VP4 at ~15-Å local resolution) arise only from VP10A at the type II channel (Fig. 6e), suggesting that the

specific shedding of spikes that occurs on partial particles may be required to form this acidified conformation.” In the Methods section, we added sentences (lines 512–516): “To prepare the grid for cryo-EM, the virus sample was first deposited onto a Lacey-carbon grid (Ted Pella, #01824) for 30 s. After removal of excess virus solution, buffer (pH 5.0) was applied onto the grid for 2 min. The grid was then loaded onto the Vitrobot Mark IV plunger for blotting for 4.5 s under 90% humidity at 16 °C before being plunge-frozen in liquid ethane.”

Accordingly, the image in Fig. 6e has been replaced with the cryo-EM structure of acidified BAV particles.

Caption:

e, Cryo-EM structure of BAV particles after acid treatment (pH 5.0), showing VP4 density as rod-shaped projections (colored in magenta) on the viral core.

We also added new Supplementary Fig. 10.

Caption:

Supplementary Fig. 10. Cryo-EM structure of acidified BAV particles.

a, A representative cryo-EM micrograph of acidified BAV particles on ultrathin carbon film. Only one 2D class (shown in the inset) was identified based on ~1,400 particles. Projections on viral particles are indicated by white arrowheads.

b, Local resolution estimation of cryo-EM reconstruction of acidified BAV particles.

Comments, questions, and suggestions:

Maybe there is a better title for the manuscript. The “stepwise detachment of viral spikes” has no physiological role in this context (e.g. preparation of core particle by alkaline treatment).

Response: We agree that saying “stepwise” lacks sufficient supporting experimental data if based only on the three atomic-resolution structures presented in the manuscript. As suggested by the referees, we have now added structural analysis of low-pH-treated virus particles as well as data on transcriptional activity of alkaline-treated cores, which we believe will be helpful to illustrate how “stepwise detachment of viral spikes” happens. Our new cryo-EM structure of acidified particles, as suggested above, verifies that removal of spikes above the type III channel is probably biologically relevant. Although we obtained cores by alkaline treatment, considering that these cores are transcriptionally active, it is reasonable to infer the removal of the VP4 protein after membrane penetration. The modified title of the manuscript is “Cryo-EM structures of Banna virus in multiple states reveal stepwise detachment of viral spikes,” which we think better describes the relationship between our cryo-EM structures. To clarify this point, we have added a sentence to the Discussion section (lines 345–349): “BAV entry into the cell is a highly ordered process with stepwise detachment of viral spikes, as reflected by the variation in spike arrangement on full particles (120 spikes containing VP4 and VP9), partial particles (60 spikes containing VP4 and VP9), acidified particles (60 spikes containing VP4), and core particles (no spikes).”

What is the consensus in the field for annotation of the Banna virus VP4 protein? In UniProt, VP4 is annotated as “outer capsid protein” (<https://www.uniprot.org/uniprotkb/Q9INI3/entry>). Here, the authors classify it as spike protein. In case it is an outer capsid protein, as UniProt suggests, shouldn't the full virion be called triple-layered particle instead of double-layered particle? Or is VP4 considered part of the VP8/VP10 layer? I agree with the authors that it should rather be assigned as spike protein.

Response: Early annotations of the structural proteins of Banna viruses were based on the work of Mohd Jaafar *et al.* (DOI: 10.1016/j.str.2004.10.017). They suggested that BAV particles contain “three distinct protein layers (subcore, core, and outer capsid) that is reminiscent of the orbiviruses and rotaviruses.” VP4 and VP9 were thought to be components of the outer capsid. This may also be the reason why the UniProt database lists VP4 as an outer capsid protein. However, on the basis of our results, it is not the case. In the full particles of Banna virus, the VP4 trimers only partially cover the viral surface area in a discontinuous way, and the “upper” domain of VP8, which locates in the outer layer of the double-layered capsid, is exposed to the external environment. We believe our findings here update understanding of the structural and functional roles of VP4 as a component of spikes, rather than being one protein in the outmost shell of a triple-layered particle. We specified this point in the first paragraph of the Discussion section: “in the double-layered full BAV particles.”

Why do VP10A and VP10B specifically intercalate between the VP8 trimers Q and R? How does this cleft structurally differ from the other interfaces between VP8 trimers around the pseudo six-folds of the channels II and III?

Response: Compared with VP8, VP2 plays a more important role in localizing VP10. The C-terminal tail of VP10A is inserted into the interface of VP2A and VP2B in the same VP2 pentamer (A2 and B1 molecules in Supplementary Fig.7b); VP10B is located between neighboring VP2 pentamers, where its C-terminus is positioned at the interface between A1 and aB. To better clarify this point, we modified a sentence at lines 202–204: “These tails are important for the interactions between VP10 and VP2, resulting in differentially orientated VP10A and VP10B within the outer capsid.”

How similar are the channels II and III, e.g. if they are superimposed?

Response: To compare the channels, we superimposed VP10B onto VP10A. The center-of-mass of the VP8 trimers surrounding the corresponding channels does not significantly shift, except that the S-trimer moves ~ 10 Å compared with the T-trimer. To clarify this, we have added a sentence in the Results section (lines 192–193): “The type II and III channels are well aligned, except for one VP8 trimer shifting ~ 10 Å (Supplementary Fig. 7c).”

We also added new Supplementary Fig. 7c.

C

Caption:

c, Structural comparison of the type II and type III channels. VP10A (magenta) and VP10B (blue) are superimposed. The hexagons formed by six surrounding VP8 trimers are shown as cartoon lines connecting the centers-of-mass of the neighboring VP8 trimers.

Spike occupancy on channel III appears to be “all or nothing” as there are particles with either 60 or 120 spikes. Do the authors have an explanation for this? Why are there no particles with partial spike occupancies? Is there density at low contour levels for partially occupied spike positions?

Response: We have no direct evidence to support the biological relevance of partial particles. However, as demonstrated by our new cryo-EM structure in acidic conditions, there is rod-like density only above the type II channel. A reasonable explanation for this is

that before the formation of the structure in acidic conditions, the spikes on the type III channel are dissociated from the virions. This observation suggests that partial spike occupancies may be biologically relevant, at least for some intermediate states during virus entry. In the partial particles, no obvious density occupies spikes above the type III channel, even when the contour level is adjusted to lower levels (see the figure below).

In the asymmetric RdRp reconstruction, are there any additional structural changes in the VP2 proteins (in addition to the N terminus) induced by RdRp binding, for instance around the five-fold pore in the apical domain of VP2A?

Response: Thank you for your question. In the asymmetric reconstruction of the fivefold vertices, the VP2s that make up the inner shell show only slight structural differences. As shown in the figure below, the individual VP2s are well superimposed (VP2A1, VP2A2, VP2A3, and VP2A4 chains are superimposed onto VP2A; VP2B1, VP2B2, VP2B3, and VP2B4 chains are superimposed onto VP2B). The RMSD values of the superimposed VP2s are all $<0.5 \text{ \AA}$ (774 C α atoms for VP2A, and 907 C α atoms for VP2B). The five VP2A molecules constituting the five-fold channel were resolved to the same number of residues (A–A4: 182–955, except one N-terminal extension shown in Fig. 2f), and there were no significant differences in the secondary structures, including the loops of the apical domains around the five-fold channel. However, in the B molecules, residues 404–428 were not resolved in the B, B1, B2, and B4 strands. Residues P404–T421 were resolved in the B3 chain due to the interaction with RdRp, which is the most significant structural variation in the B molecules. This information has been added to the caption of Supplementary Fig. 6a.

Line 233: “However, the corresponding loop region in BAV VP4 (Q319–K330) is not 234 likely to form a similar structural motif because of the presence of two destabilizing glycine 235 residues (G321 and G329) in this region (QYGNTSASEFGK) (Fig. 4e).” I’m not convinced of this argument, given that the VP5 from BTV and VP4 from BAV are paralogs, I would not be surprised if the same loop in BAV has a similar structure and function if rearranged.

Response: Owing to the lack of direct structural data illustrating this structural transformation, we decided to remove this part from the text.

Fig. 6d: The description of the conditions for lines A, B, C, and D are confusing. Were A and C treated identically (except that C was low pH-treated)? The caption says that A is “concentrated” and C is “pelleted”. The same fractions for the neutral and low-pH treated virus should be analyzed by SDS-PAGE.

Response: In the revised Fig. 6d, we have added a lane (Lane 2) to show that only a small amount of concentrated BAV particles are precipitated at neutral pH.

d

Caption:

d, SDS-PAGE analysis of the protein composition of purified BAV particles (in PBS, pH 7.4, Lanes 1–3) and BAV particles in acidic buffer (citrate–phosphate buffer, pH 5.0, Lanes 4–5). Lane 1, concentrated BAV particles; Lane 2, pelleted BAV particles after centrifugation at 10,000 × g, showing only a small amount of particles are precipitated; Lane 3, supernatant after centrifugation at 10,000 × g, showing most BAV particles are intact in solution; Lane 4, pelleted particles after centrifugation, showing most particles are precipitated; Lane 5, supernatant after centrifugation, showing VP9 protein has been released into solution.

Minor comments:

Abstract: The authors say that VP10 “plays pivotal roles in the assembly of BAV virions”, can they explain what roles? Also, why is it relevant to mention here that it has a high pI value?

Response: Theoretically, VP10 is a basic protein (pI >7) that can stabilize the viral

particle structure in neutral conditions by electrostatic interactions with acidic proteins ($pI < 7$), such as VP2 ($pI = 6.1$) and VP4 ($pI = 5.9$). Considering the limited word count allowed in the abstract, we have decided to simplify this sentence to: “Among the BAV structural proteins, VP10 is the smallest; it was identified to be a cementing protein that plays a pivotal role in the assembly of BAV virions by directly interacting with VP2 (inner capsid), VP8 (outer capsid), and VP4 (spike).”

Line 33: conformational changes in the membrane penetration protein.

Response: Modified as requested.

Line 93: “unique structural features”, please specify.

Response: Thank you for your suggestion. We have rephased “unique structural features” to “unique composition and conformation of viral spikes” to better summarize our structural data.

Line 99: nucleotide similarity (>95%), do the authors mean identity? There are similar amino acids, e.g. conservative substitutions, but what were similar bases?

Response: Thank you for your note; we have added more information and rewritten the sentences at lines 100–105: “Currently BAVs can be classified into three groups, A, B, and C, based on phylogenetic analysis of the complete coding sequence (CDS) of segment 12. The CDS of segment 12 of BAV strain YN15-126-01 has high nucleotide identity with other reported group A BAVs [YNV/01-1 (95.82%), SC043 (95.83%), and BAV-Ch (95.82%)] isolated in Yunnan, China; the amino acid identities of YN15-126-01 with YNV/01-1, SC043, and BAV-Ch are 95.65%, 96.14%, and 96.62%, respectively.”

Line 101: the host cells were inoculated with virus, not the virus. Change to “host cells were inoculated with virus”.

Response: Modified as requested.

Line 102: How is the ratio viral RNA/cells (R/C) calculated? Is it meant number of virions to number of cells? Or how is viral RNA determined? Please clarify.

Response: The ratio of viral RNA/cells (R/C) indicates the copy number of viral RNA compared with the number of cells.

The viral RNA copy number was determined by real time PCR:

(1) A viral RNA standard (125-nt) was produced *in vitro* by using a T7 High Yield RNA Transcription Kit (Vazyme).

(2) The number of copies in the standard was determined using the formula:

Number of copies = (amount of ssRNA in nanograms $\times 6.022 \times 10^{23}$) / (length of ssRNA in nucleotides $\times 1 \times 10^9 \times 330$).

(3) Real-time PCR experiments were performed using serial dilution (10-fold) of the standard RNA to obtain a standard curve: $y = -3.807x + 52.23$, $R^2 = 0.9991$, where y is the Ct value, and x is the log of the RNA copy number.

(4) The viral RNA extracted from the BAV was tested by real-time PCR, and compared with the standard curve to determine the copy number.

For more detailed information on qRT-PCR, please refer to the paper by Gibson *et al.* (DOI: 10.1101/gr.6.10.995).

One sentence has been modified in the Results section (lines 106–109): “To evaluate the infectivity of BAV YN15-126-01 in a variety of host cells, six vertebrate cell lines (LMH, BHK-21, PK15, Vero, A549, Huh-7) and one mosquito cell line (C6/36) were inoculated with virus at a ratio of copy number of viral RNA to number of cells (R/C) = 1 or 1000.”

Two sentences have been added to the Methods section (lines 378–388): “A viral RNA standard (125-nt) was produced by using an *in vitro* T7 High Yield RNA Transcription Kit (Vazyme) and serial dilutions (10-fold) of the standard RNA were used to generate a standard curve for RT-qPCR.” “The viral RNA copy number (copies/ml) in cell culture was calculated based on the standard curve [$y = -3.807x + 52.23$, $R^2 = 0.9991$, where y is the Ct (cycle threshold) value, and x is the log of the RNA copy number].”

Line 118, 120: not an electron density map, Coulomb potential or just density map.

Response: Modified as requested.

Line 131: Why is it “pseudo T = 2” and not just “T = 2” lattice?

Response: The ‘triangulation number’ (T) of the icosahedral symmetry follows equation $T = h^2 + hk + k^2$, where h and k are integers, as first defined by Caspar and Klug (DOI: 10.1101/sqb.1962.027.001.005). On the basis of this equation, T cannot be 2 (Rossmann, DOI:10.1017/S0033583513000012), and “pseudo T = 2” has been used to describe the inner capsid protein of mud crab reovirus (Zhang *et al.*, DOI: 10.1371/journal.ppat.1011341).

Line 165: DALI server, please add citation.

Response: Modified as requested.

Line 168: Citation, S. Jenni, et al., In situ Structure of Rotavirus VP1 RNA-Dependent RNA Polymerase, Journal of Molecular Biology, <https://doi.org/10.1016/j.jmb.2019.06.016> could be added here, as not all N-terminal VP2 tails interacting with the rotavirus RdRp were modeled in reference 25.

Response: Modified as requested.

Line 169: Delete “Unexpectedly”.

Response: Modified as requested.

Line 299: “more phylogenetically related to orbivirus BTV” more than which other virus?

Response: The “other virus” refers to rotavirus (RRV), which is one of most studied sedoreoviruses. The DALI results suggested that the RdRp structure of BAV is more closely related to that of BTV (Z-score = 17.5) than to that of RRV (Z-score = 16.8) (Supplementary Fig. 6f). In addition, BAV has a penetration protein (VP4) very similar to the penetration protein of BTV (VP5), which are different from the counterpart in RRV (VP5*). To make this clearer, we modified this sentence to, “BAV is probably more closely phylogenetically related to orbivirus BTV than to rotavirus RRV.”

Line 312: replace “correctly”, with “as inferred from paralogs”.

Response: Modified as requested.

Line 384: what are Ct values?

Response: In a real-time PCR assay, a positive reaction is detected by accumulation of a fluorescent signal. The Ct (cycle threshold) value is defined as the number of cycles required for the fluorescent signal to exceed the background level. This information has been added to the Methods section.

Line 648: Please remove the statement that the map was contoured at 1.6 sigma. The value of 1.6 is meaningless, as it depends on the size of the box of the reconstruction and the masking of the density. Meaningful would be at what absolute value of Coulomb potential the map was contoured, or at what absolute density level (e.g. protons per cubic Angstrom).

Response: Modified as requested.

Line 650: Please define ASU (asymmetric unit).

Response: Modified as requested.

Line 659: doesn't look like map was filtered at 6 angstrom resolution, looks like one can see individual atoms.

Response: Thank you. The map is actually presented as a solvent-excluded surface. We modified “filtered at 6-Å resolution” to “presented as solvent-excluded surfaces with a probe radius of 1.4 Å.”

Line 688: five types of VP8 trimers

Response: Modified as requested.

Line 690: three types of channels

Response: Modified as requested.

Lines 751 and 752: color description of domains, needs editing?

Response: Thank you. We re-edited this sentence.

Fig. S1b: What is dpi, please define?

Response: “dpi” means “days post-infection” and this information has been clarified in Fig. S1b.

Caption Fig. S1b: What is CPE, please define?

Response: CPE means “cytopathic effect,” and this information has been added to the caption of Fig. S1b.

Fig. S8c: The structure of the Banna virus VP4 protomer should be added next to the BTV VP5 protomer in the same representation for direct comparison.

Response: As requested, we have moved the original Supplementary Fig. 8c to be Fig. 4e.

Caption Fig. 5a: analysis of alkaline-treated core particles, is this after purification of core particles, e.g. by pelleting?

Response: The alkaline-treated core particles were purified by procedures described in the Methods section (also shown in the right panel of the original Fig. 5b) before the SDS-PAGE analysis. To better present the results, we have switched the order of Fig. 5a and Fig. 5b.

Caption Fig. S9, line 138: Replace “Dynamics” with “Structures”

Response: Modified as requested.

Fig. S8e. Why are the VP2 and VP8 bands much fainter in lane C (virions incubated with 400 mM MgCl₂ in the absence of trypsin) compared to the control samples in lane A or B?

Response: We repeated the experiment with different concentrations of MgCl₂ (see the figure below). We found that if the concentration of MgCl₂ was ≥ 400 mM, the VP2 band was decreased more significantly than other virus components (left panel). However, if we adjusted the pH value of the mixture with 200 mM Tris-HCl (pH 7.4), VP2 and other viral components remained unchanged in the solution (right panel). Therefore, variation in pH after adding a high concentration of MgCl₂ may lead to the decreased VP2 on the

SDS-PAGE gel. Given the low transcriptional activity of the core produced in our enzyme digestion experiment (see above), we decided to remove the original SDS-PAGE images and TEM images (Supplementary Fig. 8e, f) from the revised manuscript.

References:

1. Mohd Jaafar F, Attoui H, Bahar MW, Siebold C, Sutton G, Mertens PP, et al. The structure and function of the outer coat protein VP9 of Banna virus. *Structure*. 2005;13(1):17-28.
2. Mohd Jaafar F, Attoui H, Mertens PPC, de Micco P, de Lamballerie X. Structural organization of an encephalitic human isolate of Banna virus (genus Seadornavirus, family Reoviridae). *J Gen Virol*. 2005;86(Pt 4):1147-57.
3. Gibson UE, Heid CA, Williams PM. A novel method for real time quantitative RT-PCR. *Genome Res*. 1996;6(10):995-1001.
4. Caspar DL, Klug A. Physical principles in the construction of regular viruses. *Cold Spring Harb Symp Quant Biol*. 1962;27:1-24.
5. Rossmann MG. Structure of viruses: a short history. *Q Rev Biophys*. 2013;46(2):133-80.
6. Zhang Q, Gao Y, Baker ML, Liu S, Jia X, Xu H, et al. The structure of a 12-segmented dsRNA reovirus: New insights into capsid stabilization and organization. *PLoS Pathog*. 2023;19(4):e1011341.

Reviewers' Comments:

Reviewer #1:

Remarks to the Author:

I am fully satisfied with the responses to my comments by the authors and also the revisions that are made.

Reviewer #2:

Remarks to the Author:

I appreciate how the authors have revised the manuscript based on the reviewers' comments. In response to the newly added cryo-EM reconstruction obtained from low pH-treated BAV particles, I have two more suggestions:

I think it's fair to conclude that the spikes observed in the low-resolution reconstruction of BAV particles after acid treatment are VP4, given its similarity to bluetongue virus VP5 and the results reported in reference 11. Can the authors briefly discuss how their reconstruction, and specifically the conformation of VP4, relates to the observations in reference 11?

I think the VP4 spikes in the density map of BAV particles after acid treatment (Fig. 6e) should be colored in beige (instead of magenta), as in the other figures.

Otherwise, I have no reservations in recommending the work for publication.

In response to Referee #1:

Reviewer #1 (Remarks to the Author):

I am fully satisfied with the responses to my comments by the authors and also the revisions that are made.

Response: Thank you.

In response to Referee #2:

Reviewer #2 (Remarks to the Author):

I appreciate how the authors have revised the manuscript based on the reviewers' comments. In response to the newly added cryo-EM reconstruction obtained from low pH-treated BAV particles, I have two more suggestions:

I think it's fair to conclude that the spikes observed in the low-resolution reconstruction of BAV particles after acid treatment are VP4, given its similarity to bluetongue virus VP5 and the results reported in reference 11. Can the authors briefly discuss how their reconstruction, and specifically the conformation of VP4, relates to the observations in reference 11?

Response: As suggested by the referee, we have added a sentence at the end of the first paragraph in the Discussion section: "Indeed, on acidified particles, the rod-shaped projections of VP4 are reminiscent of the six-helix stalks formed by BTV VP5¹¹."

I think the VP4 spikes in the density map of BAV particles after acid treatment (Fig. 6e) should be colored in beige (instead of magenta), as in the other figures.

Response: Modified as requested.

Otherwise, I have no reservations in recommending the work for publication.

Response: Thank you.